

# Visual log analysis method for designing individual and organizational BIM skill

Tsukasa ISHIZAWA*[1] and Yasushi IKEDA*[2]

[1] Ph.D. Candidate, Graduate School of Media and Governance, Keio University
  Group Leader, Computational Design Group, Advanced Design Department,
  Takenaka Corporation
[2] Professor, Graduate School of Media and Governance, Keio University

* tucasa@sfc.keio.ac.jp

## Abstract

Extending the existing Building Information Modeling (BIM) log mining approach, the paper proposes a novel method for visual analytics on clustered logs collected from multiple organizations. Implementing multiple datasets as case study processes demonstrated that the analytics effectively visualized the structured BIM events in command, organization, and user layers. Identified four major cluster groups correspond to logs' commonality and level of contribution significantly helped decipher recorded BIM activities without requiring background knowledge. The proposed technique allows the intercomparison of BIM activity that enables data-driven skill design for individuals and organizations. Such monitoring allows BIM users and teams to respond to transient project situations dynamically, which is expected to mitigate the observed major BIM problems of skill and management. The method contributes to increasing the likelihood that the model will be used throughout the project duration, leading to enhanced BIM's expected value in projects.

## Keywords

Building Information Modeling, log mining, visual analytics, skill design, intercomparison, self-monitoring

## 1.  Background

### 1.1. The missing piece to complete the BIM lifecycle

Many countries have mandated the use of building information modeling (BIM) in domestic building and construction projects [1,2]. In particular, these governments anticipate that the use of BIM will grow industrial productivity, which over the past half-century has stagnated [3]. Furthermore, increasing demand for greener or more environmentally efficient buildings has helped increase demand for integrating BIM into design processes [4]. BIM has become an indispensable tool for today's large-scale building projects, with little doubt remaining concerning its significance in the design and construction process.

 Despite its obvious necessity, some studies have found that, in actuality, BIM use is often abandoned in the project. The difficulty in sustaining BIM use mainly lies on the bridge from design to construction and facility management. Eadie et al. found that BIM is most often used in the early stages of the project, such as the design and preconstruction stages (i.e., detailed design and tender stages), with use progressively diminishing with project development into later stages such as in the feasibility, construction and operation and management stages [5]. This focus on BIM in the early stages of a project is similarly found in academia. Sacks et al. highlighted that the majority of academic and industrial research on BIM focuses on design and preconstruction planning with significantly lower efforts taken towards the development of BIM-based tools to support coherent production management on-site [6].

 In addition to encouraging BIM in construction and subsequent stages of the project, linking BIM between design and construction is ever more seen as a critical step in completing the

**Type**: Research article

**Citation**:

**Received**:
**Revised**:
**Accepted**:
**Published**:

project BIM lifecycle. The United Kingdom includes such an outlook in its BIM strategic plan (Digital Built Britain). The Singapore government launched an implementation plan for Integrated Digital Delivery (IDD) to integrate building processes and stakeholders, including digital design, digital fabrication, digital construction, and digital asset delivery and management [7]. Raising the likeliness of maintaining the BIM initiated in the design stage to the end of the project leads to achieving these industrial goals.

## 1.2. Managing a cross-functional BIM team

Chien et al. identified thirteen risk factors in BIM implementation related to the technical, management, personnel, financial, and legal aspects of the implementation, with the management of BIM being a primary risk factor in BIM adoption. Among these factors, four factors are related to the transition of management and workflow: F3 (challenges in model management), F5 (challenges in management process changes), F6 (inadequate upper management commitment), and F7 (challenges in workflow transition) [8]. In particular, management's strong commitment is indispensable in avoiding or reducing the associated risks in BIM projects.

Alternatively, Fazli et al. described that project managers, in general, have minimal knowledge of BIM and are usually unappreciative of the benefits of BIM, especially in construction projects. Furthermore, BIM is still perceived as a newly required skill in the industry, and learning BIM demands a steep learning curve [9,10]. BIM projects are often overseen by managers who have only limited knowledge of BIM or adhere to the conventional drawing-oriented workflow. Attempts to change this situation all at once may prolong these problems even further.

Efforts have been made to mitigate such a mismatch. Several studies have proposed performance monitoring systems to determine the actual BIM activities undertaken in a project. The systems measure and visualize how BIM specialists behave, which facilitates BIM managers in improving BIM productivity and in defining an appropriate staffing strategy. Such monitoring systems may potentially become a self-diagnostic system for the project. Bryde states that BIM can be a catalyst for project managers to reengineer their process to integrate better the different stakeholders involved in modern construction projects [11].

Recent research has found that BIM log mining is practical for investigating productivity in BIM. BIM log mining applies an analytical approach to analyze the command history automatically recorded by BIM software. It provides the opportunity to discover implicit patterns in use without requiring extra effort from the BIM users. BIM log mining is expected to increase BIM performance and enhance model communication. This study seeks to further contribute to the potential of BIM log mining by proposing a novel approach to mine the BIM logs.

## 1.3. Defining the research area

Rectifying conflicts that result from faulty design decisions account for 5 to 8% of total project costs [12]. A mean design error cost of 14.2% of a project's contract value has been reported [13]. Design errors occupy a notable portion of project costs, especially considering that the design process itself typically accounts for approximately 5 to 10% of the total cost [14]. As a result, the cost of rectifying design errors is nearly the same magnitude as the modeling process in the design phase.

As the model proceeds along its lifecycle, especially in the construction phase, BIM teams become multidisciplinary and cross-functional [15–18]. In such an environment, known as a big room concept [19], the individual BIM specialists update their respective models to complete the project model as a single, trusted information source for the project. These specialists are responsible for creating individual models and the model federation, clash coordination, and documentation, which occupy a considerable amount of time. Managing team performance and enhancing collaboration requires a multidimensional assessment, which is impossible to perform using the modeling-based performance monitoring systems.

Due to the nature of strictly classified BIM components, similar operations are often undertaken by different individual commands associated with the object category. For example, Autodesk Revit has tens of different commands that can be used to erect walls in the model. There exist four different commands to create a straight wall depending on whether it is a structural wall or not and refers to the existing linear geometry. To paste objects from the clipboard, a user has six choices of commands per the object alignment. It is known that the general commands are most frequently used as compared to the function-specific commands [20]. Preceding research has mainly focused on the most frequently used commands collected from BIM logs; however, less frequently used commands may need additional attention if they

have significance to the project. Such less frequent yet relevant data entry, known as weaker signals in the data mining field, require specific processes to be equally treated with the very frequently issued commands.

## 2.  Research question and expected contribution

Steady modeling progress and design option exploration are vital in the early stages of design. As the project comes to the construction phase, numerous amendments and revisions are required besides the model creation. The proportion of drawings generally increases as projects progress.

Various formations are possible to supply the BIM workforce with a project. A group of BIM-enabled specialists may fluently converse over models; analyzing the frequency of model use should enhance BIM communication. The intensiveness of model-making especially matters when specialized personnel like BIM operators undertake BIM tasks; in this case, engagement of other project members is vital to leverage the building information for projects.

The relationship between diverse organizations and the range of functionality BIM software provides dynamically transforms depending on the collaboration structure or project requirement. Stakeholders increases in large projects, and thus the need arises to mediate more complex connections.

The interpretation of recorded BIM activity facilitates reviewing the dynamically changing collaboration formation. For example, creating walls is one of the typical commands in modeling software. The frequency and context of this specific command imply the situation of BIM use. However, considering that the provided software commands exceed a thousand, a full-scale analysis will end up feeding overwhelming information, thus not useful in practice. Alternatively, an unsupervised machine learning algorithm partitions the log data by similarity avails for simplicity. Moreover, obtaining the typology and commonality of a series of software events is universally applicable, regardless of organization, team, and project types.

The key to BIM implementation is to design the appropriate skill-set and organization required for various missions. This paper proposes introducing clustering and visual analytics to the BIM log mining method, an emerging process mining approach in BIM, to decode the recorded BIM event and propose a process for overviewing BIM activities independent of professions or disciplines. Past research has established that BIM log mining helps improve individual skills and staffing strategies. This study extends the system to enable the intercomparison of characteristics in BIM use across projects, organizations, and users. Self-monitoring of the status of BIM's effect by the method contributes to improving the skill design of individuals and organizations.

## 3.  Literature review

Log mining, or log analysis in the broader context, is a technique that leverages data mining for the analysis of logs, which are records related to the activities occurring on a system. We expect log mining to gain situational awareness, discover new threats, and extract actionable findings automatically. Though the technique can be applied across industries, it is particularly known in specific fields, for instance, web-based systems known as web log mining [21–23]. The common purposes of log mining are anomaly detection [24], or result optimization [25]. There also studies that have suggested increased productivity from log mining. For example, in the automotive industry, Niimi et al. succeeded in extracting the tacit operational skill of 3D CAD modelers in vehicle design by analyzing operation logs [26].

In recent years, several studies have emerged that applied the technique to the design log of BIM software. The first application was reported by Yarmohhamadi et al. as the pattern mining methodology to extract sequences commonly shared among users [27]. This study proved that the unstructured BIM log files from Autodesk Revit could transform into Generalized Suffix Trees (GST) data structures through appropriate data treatment and that meaningful information can be derived by the analysis of the dataset. The subsequent research further investigated the presence of implicit patterns in log files and empirically characterized the performance modelers [28]. Zhang et al. proposed a pattern retrieval algorithm to discover typical design command sequences [29]. Those reports mainly addressed the command names issued during the operational sessions. The primary intention of these pieces of research was to establish the monitoring process of modelers' performance for achieving higher design productivity and effective management. However, as briefly stated by Zhang, the measurement of design productivity is much more difficult than that of construction productivity. Thus, he emphasized that the proposed metric should be interpreted in the context of other performance measures for the project instead of replacing or undermining other performance metrics to

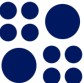

assess design projects [29]. Pan et al. classified designers into three clusters using a clustering method called node2vec-GMM, explaining that the three clusters have different characteristics and understanding them can provide data-driven support for organizational management. This study has already explicated the utility of clustering for BIM event logs. In this study, the authors aim to expand the dataset collected from multiple organizations and apply visual analytics to empower mining with extended scalability [30].

These BIM log mining methods are strongly inspired by past research that has been treated in the area of business process management. The significant past contributions include Aalst's publication, which holistically presents various process mining techniques that aids organizations uncover their business processes [31]. Sequence mining is another domain that addresses the pattern discovery problem. Past contributions include the paper by Wang et al., which presents an efficient algorithm to mine frequent closed sequences without candidate maintenance [32]. The visualization of event sequence data influences the application of visual analytics. The essential existing research includes the paper by Du et al., which describes 15 strategies of temporal event sequence analytics grouped mainly as 1) extraction strategies, 2) temporal folding, 3) pattern simplification strategies, and 4) iterative strategies [33].

BIM projects are often large and complex, where internal collaboration can significantly influence overall productivity. Zhang et al. applied the social network model to analyze BIM designers' characteristics and social performance in an architectural firm. The research shed light on the management issues to proffer the system for the appropriate staffing strategy, the assignment of the right design tasks to work on, and the discovery of bottlenecks in the design process development [34].

Past research has made a substantial contribution to BIM management. Tauriainen et al. analyzed design management issues and identified the major causes of the problems, including the insufficient BIM knowledge and experience of BIM managers [35]. Barison et al. outlined the areas of responsibility of multiple BIM specialists [36]. Uhm et al. categorized BIM roles into eight categories through the analysis of job postings and analyzed the relationships between them [37]. Kassem et al. identified the core competencies of four key BIM specialist roles, including the BIM Manager and the Information Manager, revealing substantial overlap across all roles [38]. The study regarding BIM team and talent management increases its necessity. The 2013 McGraw Hill SmartMarket Report stated that most contractors encourage BIM expertise in team formation [39]. Taking the increasing demand into account, it is evident that today, BIM ability has a greater influence over the startup of a project team. Davies et al. identified important soft skills, such as communication, conflict management, negotiation, teamwork, and leadership, for BIM project teams that are generally overlooked [40]. Davies et al.'s work implies that such skill developments are necessary to move forward from the implementation stage to the leveraging stage.

The effort has been made to develop measurement methods for the model use and level of implementation [41–44]. Wu et al. exhaustively overviewed nine mainstream BIM measurement tools and discovered that those tools have a unique emphasis on understanding users [45]. Most BIM measurement tools have organization or human-related measurement features. These methods mainly evaluate the use of BIM-related tools, the availability of key personnel, the provision of education programs, or stakeholders' satisfaction.

The aforementioned literature supports the conclusion that the magnitude of each aspect of BIM, such as experience, cost, workflow, or contracts, are almost equal to each other. As briefly summarized by Miettinen et al., there is no single satisfactory definition of BIM. Rather, it needs to be analyzed as a multidimensional, historically evolving, complex modeling method [46].

BIM log mining was first intended to be used in design productivity by focusing on frequently-issued modeling commands. Its target use gradually expanded to cover other factors, such as team collaboration or modeling strategies. A BIM team communicates with more stakeholders as the project progresses. The BIM tasks on such project stages are widely varied and may include non-modeling-related activities as well. Therefore, BIM log mining that flexibly analyzes cross-functional BIM activities based on the current requirement is a novel approach that contributes to the body of knowledge and provides a valuable tool for improved BIM project management.

## 4. Methodology

This paper is part of a doctoral research investigation following the Design Science Research Methodology (DSRM) for information system research [33,47]. The six steps in the methodology are shown in Figure 1. This paper focuses on the problem identification steps through evaluation referring to the existing BIM log mining approaches.

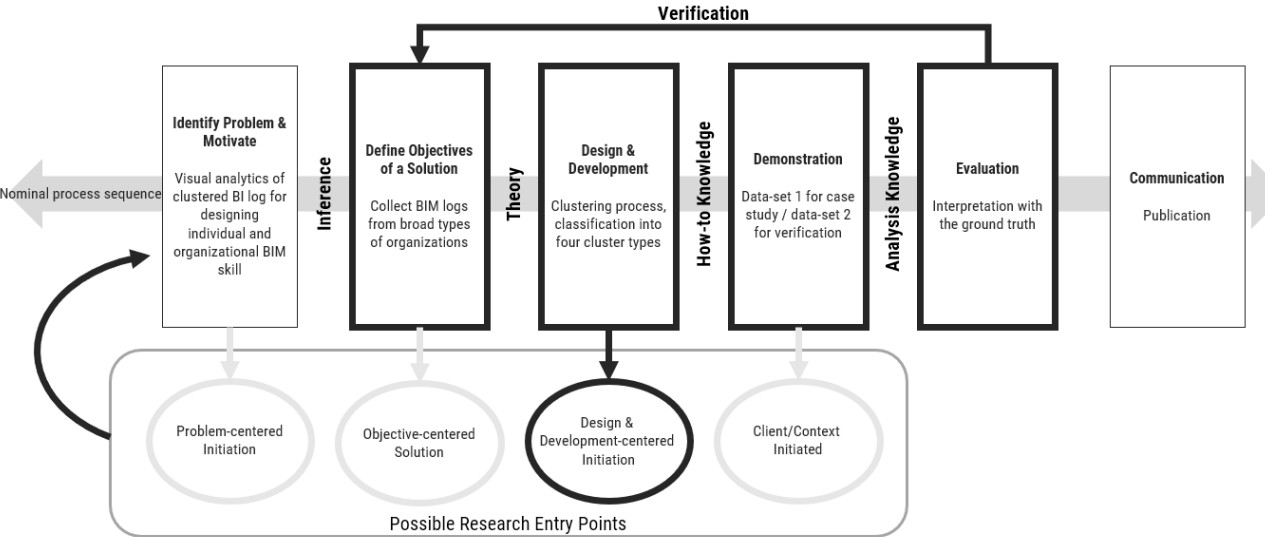

Figure 1. Proposed research methodology based on design science research methodology (DSRM) for information system research.

The research and development proceed with five steps. First, the authors collected BIM logs from various organizations to compose two datasets for case study purposes. Second, an algorithm was developed that transforms the collected log data into comma-separated-value (CSV) format, and the clustering algorithm was applied to partition the logs into a limited number of clusters. Third, the visual analyses are carried out on datasets in three different contexts based on command, organization, and user. The obtained clusters were grouped into four types by the observed characteristics. Fourth, the evaluation was made on the clusters and their types to interpret the log files. Last, the another case study further tested the process and the utility.

## 5. Process

### 5.1. Data collection

The subject of the analysis was the event logs, called journals, produced by Autodesk Revit. A single journal file captures the actions taken by the software from the beginning to the end of the software session. Though the journal was initially intended to troubleshoot technical problems with the software [48], past studies proved it serves the process mining purpose as it contains rich enough information to track the details of user activities. Moreover, as Revit is a globally adopted BIM authoring software, journal files are ideal for collecting information from multiple disciplines and regions. The records were collected to construct two datasets for the case study purpose. Table 1 presents the details about the respective datasets.

Table 1. Details of the collected datasets.

|  | Dataset 1 | Dataset 2 |
|---|---|---|
| Number of organizations | 10 | 2 |
| Number of data donors | 37 | 182 |
| Disciplinary of data donors | Various | Various |
| Number of collected log files | 1,299 | 8,432 |
| Issued commands (effective) | 226,383 | 565,289 |
| Issued commands (overall) | 622,793 | 1,113,977 |
| Average issued commands per log | 479 | 132 |
| Collection Period | February 2018 – May 2018 | April 2019 – August 2019 |
| Method | Manually transferred | Automatically transferred |

The events for Dataset 1 were collected from thirty-seven data donors belong to ten organizations. Each data donor manually sent the log files via email or file transfer services from February to May 2018. Overall, 1,299 files were collected, which contained 622,793 commands. Past studies have probed between 181,969 to 620,492 commands, and thus a dataset of this size was targeted to draw parallels to the existing literature [27–29]. Table 2

describes the details of the submitted log files, the donors, and the organizations involved. Besides architecture firms and general contractors, the donor organizations included a university, a BIM operator firm, and BIM consultants. The general contractors labeled A, H, J are the international branch offices of F. All firm provides design-build service including the engineering of structure, MEP and construction. Though their BIM workflows generally follow the same principle, the details are individually interpreted, reflecting the local collaboration schema. BIM operator company E exclusively serves for F to outsource the CAD and BIM workforce. The BIM consultants provide a similar service. Company G additionally provides the strategic planning for BIM implementation and advancement. The data donors from organization D were the students taking an architectural modeling course. The project BIM models were not collected for confidentiality reasons.

The authors additionally carried out a user survey about the data donors in August 2018 to acquire the donors' BIM skills and project status. Twenty-six out of the thirty-seven data donors answered twenty questions, detailed in the Appendix. Table 2 additionally includes a short description of the donors' role, as understood by this survey.

Dataset 2 is separately constructed. More extensive log files were provided by two organizations (Label E and F in Table 2). The data acquisition took place between April and August 2019; therefore, no overlap exists between the two datasets. One hundred eighty-two architects and engineers consented to share their logs. An automation program was installed on the individual computers to transfer the log files to the server periodically.

Table 2. Details of organizations and data donors, datasets for the case study.

| Label | Type of Organization | Country | # of Data Donors | Data Donor's Role | # of Log Files |
|---|---|---|---|---|---|
| A | General Contractor | China | 2 | To audit received design models for construction study | 37 |
| B | Architect Firm | France | 1 | To audit and amend multiple design models by other in-house BIM staff | 12 |
| C | BIM Consultant | Japan | 1 | To coordinate received design models | 73 |
| D | University | Japan | 3 | To create models for design studio | 46 |
| E | BIM Operator Firm | Japan | 10 | To create and amend design models as per instruction by organization F | 272 |
| F | General Contractor | Japan | 4 | To create, coordinate and issue design and construction model | 110 |
| G | BIM Consultant | Singapore | 8 | To support BIM process | 264 |
| H | General Contractor | Singapore | 5 | To develop construction models from received CAD design drawings | 355 |
| I | Architect Firm | Singapore | 1 | To create design models for projects | 11 |
| J | General Contractor | Slovakia | 2 | To create design models based on CAD drawings by in-house architects | 119 |
| Total | | | 37 | | 1,299 |

## 5.2. Development of the algorithm

### 5.2.1. Assembling database

The classification aimed to separate the log files based on the types of issued commands and their frequency during a single software session. The clustering process requires the structured aggregation of the issued command types per user. The authors tailored a series of Python scripts to parse the unstructured text in the collected log files. The results were accumulated in a Comma Separated Value (CSV) format for the subsequent analysis.

First, the Python code reads lines from a series of log files to extract the unique IDs of issued commands triggered by the tag "Jrn.Command". The extracted IDs, as a command sequence per log, were added into a single CSV file.

Second, the invalid command "ID_CANCEL_EDITOR" was omitted from the sequence. Every time the user presses the escape key, this specific command is issued to cancel their current command mode. Previous studies have also excluded it [27–29]. Consequently, the whole log database consisted of 798 different commands, accounting for 48% of all Revit commands.

Third, a term-weighting factor was optionally applied to the database to enhance the clustering result. Some commands are repeatedly and commonly issued, such as ID_EDIT_MOVE (to move objects) or ID_ALIGN (to align the object to other selection), and appear in most logs. These commands are too generic to surmise the activity context. Additionally, as users often issue them hundreds of times more than other commands, they can strongly influence the clustering estimator resulting in the degradation of the discriminability

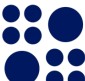

of the machine learning process. Term frequency-inverse document frequency (tf-idf) is a universal numerical statistic that reflects the importance of the terms in a corpus that are often used for information retrieval and text mining [49]. After applying tf-idf, the counts of general commands are offset and commands uniquely found in specific users increase proportionally.

### 5.2.2 Clustering

There are two types of algorithms in machine learning, supervised and unsupervised. Since no ground truth for the grouping was established beforehand, the estimator was selected from unsupervised methods. KMeans [50] and the Variational Bayesian Gaussian Mixture Model (VBGMM)[51] were the most feasible algorithms to solve this clustering problem.

The performance of the discriminability should select the algorithm. Silhouette is a method for evaluating the consistency of clusters of data in the partitioning problem. The silhouette analysis allows evaluating the methods that create better tightness and separation. Values range from -1 to 1; the closer the value is to 1, the sample displays a better match to its own cluster. When the value is near 0, the sample is close to the boundary of the clusters. The value closer to -1 offers that the sample may be misassigned to another cluster [52].

Table 3 shows the silhouette coefficient values for the result of partitioning Dataset 1 into 30 clusters, testing for each of KMeans and VBGMM with and without tf-idf application. Since the maximum value among the combinations represents the most successful separation, VBGMM without Tf-idf was selected as the best estimator. This choice is consistent with the fact that the K-means method implicitly assumes hyperspherical clusters in shape and numbers of objects in clusters are equal, so it is challenging to extract structures that violate this assumption. As shown in the following sections, the cluster sizes broadly diverse, and they reflect distinct characteristics.

Table 3. Silhouette analysis for algorithm selection.

|  | K-means | VBGMM |
| --- | --- | --- |
| Tf-idf applied | 0.191 | -0.059 |
| Tf-idf not applied | 0.146 | 0.308 |

Table 4. Silhouette analysis for determining the number of clusters.

| # of clusters | Silhouette coefficient |
| --- | --- |
| 15 | 0.306 |
| 20 | 0.316 |
| 25 | 0.309 |
| 30 | 0.322 |
| 35 | 0.34 |
| 40 | 0.322 |

Although the algorithm requires determining the number of clusters to partition, this value is non-deterministic. As an exploration, the silhouette coefficient is applied to experiment with the different numbers of clusters, whose list is shown in Table 4. Considering that the randomness of the initial values affects the clustering results, no prominent tendency was observed. Too few clusters seemed not to yield the desirably discriminating results for a broad range of data sources. Therefore, we hypothetically employed 30 clusters and attempted to interpret the results through visual analytics by incorporating the user survey.

### 5.2.3. Implementation and interpretation of obtained clusters

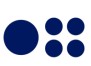

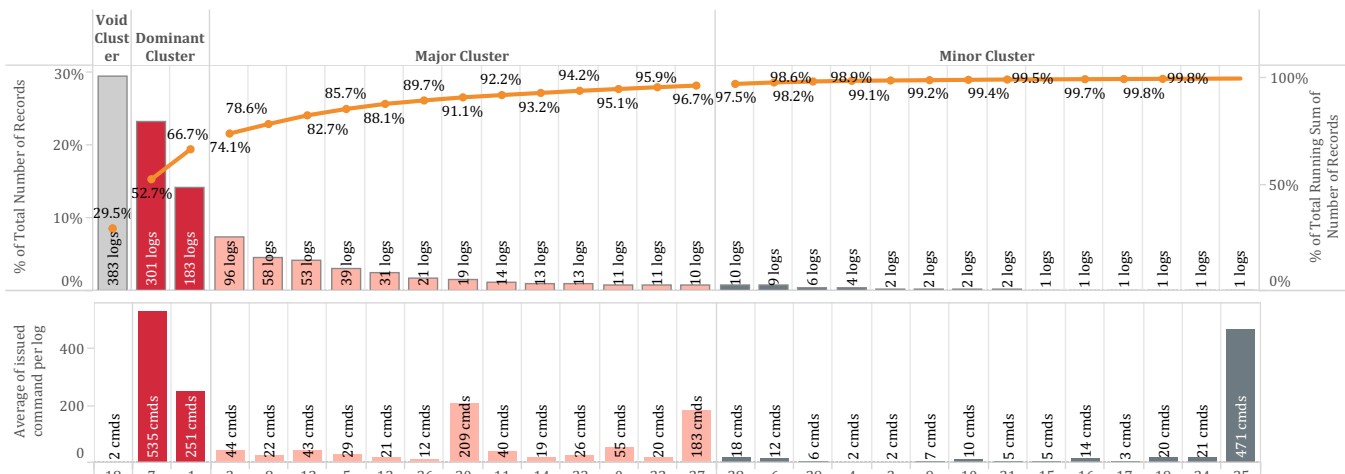

Figure 2. Overview of predicted clusters with four cluster types, dataset 1.

Figure 2 summarizes the clustering result for Dataset 1. Two different indices are expected to measure the cluster size: the number of log files the cluster contains and the average number of executed commands in those logs. Clusters with a high number of log files indicate that they are observed more frequently, and the average number of commands is relevant to the users' activeness, thus approximately indicate the level of contribution to the model.

The separation results, viewed from the number of log files in clusters, are distributed in a long tail fashion. On the other hand, clusters #7, #25, #1, #20, and #27 stand out in terms of the average number of events, while the rest have less than a few dozen records overall. Tens of commands are practically very few for substantive modeling contributions to project BIM, like creating architectural elements and drawing annotations. Therefore, a log with a small number of average command execution implies low engagement of users with model progression but rather in terms of, for example, checking content and interacting with other environments.

The distribution of logs and executed commands offers four major tendencies to which the obtained clusters can be further grouped. The first is the Void cluster; the very distinctive cluster #18 forms a group by itself. It is the largest cluster, accounting for 29.5% of the total files, but with an average event count of only two. These extremely few operations imply that a user opens the model file and exit the software with almost no action. Possible situations include opening the wrong file, checking the model version, or quitting it without knowing how to navigate.

The second is Dominant clusters that account for 66.7%, together with Void clusters. The logs under this group have characteristically high average numbers of recorded events. This type considerably frequently appears, indicating that a user performed intensive operation during the software session. It is reasonable to assume that this log signifies the main contribution to creating and editing the model.

Finally, close numbers of clusters belong to groups when the remainder is split at the double standard deviation in the running total, respectively termed as Major and Minor clusters. Both cases do not present high occurrence, and the average command issuances are also not as massive as in the Dominant cluster.

In the subsequent sections, the clusters and their groupings are interpreted by several visual analysis approaches.

## 5.3 Evaluation

### 5.3.1 Command-based analysis

The functionality of issued commands per log and cluster plays a vital role in comprehending the result of an unsupervised machine learning algorithm. The 798 different commands contained in records can be categorized into seven by their purpose: i) modeling commands (to generate components), ii) drawing commands (to draw or annotate in two-dimensional), iii) editing commands (to edit existing elements), iv) save commands (to save or synchronize current file), v) imp/exp commands (to import, export, link and manage relevant files), vi) workset configuration (to set or activate workset) and others as vii) miscellaneous. Figure 3 illustrates the ratio of each command function category per cluster.

Progressing modeling is an elementary contribution to BIM throughout the entire project

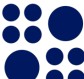

lifecycle. The dominant cluster has the highest proportion of modeling commands, and many clusters in the major cluster also display a high ratio, most notably cluster #20 as the extreme. Adversely, the clusters under Minor and Void cluster groups do not have such a feature. Drawings are still paramount in most BIM workflows. Interestingly, drawing commands are prevalent in minor clusters, particularly in clusters #6, #10, and #16. Imp/exp commands and workset configuration are typically used in integrating multidisciplinary models or site models; thus, single-model projects seldom use this functionality. Therefore, log files characterized by these commands exhibit the active collaboration occurring in the BIM environment. These commands scarcely appear in Dominant clusters and are more common in Major and Minor clusters. It implies the users are more responsible for collaboration than the modeling itself, such as model checking, coordinating, and outputting.

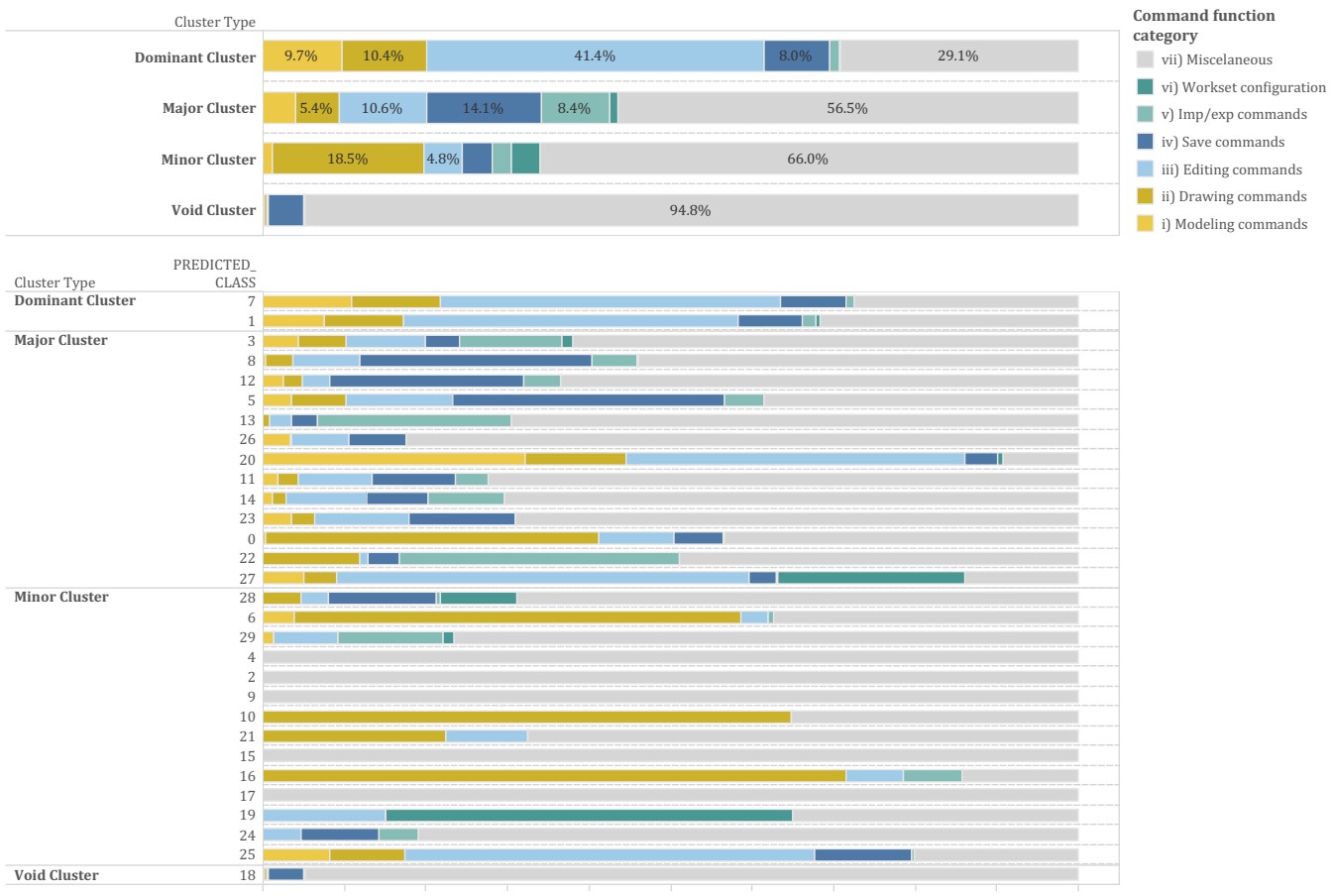

Figure 3. Command-based visualization of predicted clusters.

Whereas the proportion of command categories portrays the clusters under dominant and major cluster groups, minor clusters present distinctive distributions. For example, the miscellaneous command category entirely occupies clusters #4, 2, 9, 15, and 17. Drill down to the individual commands will further uncover the mean contributions in those records. The top three most frequently-issued commands in minor clusters were listed in Table 5.

The top command in cluster #6 is to measure the distance between objects. This command's repeated use without accompanying modeling or drawing command outlines the purposes such as CAD-based drafting or manual quantity takeoff. Cluster #2 consists only of commands to open files and a visual programming environment termed Dynamo; the direct entry to this mode exhibits the exclusive involvement in computational design scripting, including automation or optimization. Cluster #15's commands for photorealistic rendering straightforwardly mean that the presentation image was produced in Revit. The above offers that minor clusters are not positively associated with the modeling or drawing production, but they alternatively make advanced contributions by leveraging specialized features.

Table 5. Top 3 most issued commands per cluster in Minor Cluster Group, rank # is among overall dataset 1.

| Cluster # | Command Function | Rank | Command Function | Rank | Command Function | Rank |
|---|---|---|---|---|---|---|
| 28 | Configure partitions | 106 | Relinquish model ownership | 120 | Open Revit file | 16 |
| 6 | Measuring object distance | 32 | Switch to default 3D view | 10 | Open Revit file | 16 |
| 29 | Create new file with template | 172 | Quit Application | 39 | Configure units | 191 |
| 4 | Open most recently used Revit file | 91 | Configure model material | 146 | | |
| 2 | Visual programming environment | 268 | Create new file with template | 172 | | |
| 9 | Render image file using cloud | 213 | Open Revit file | 16 | Quit application | 39 |
| 10 | Configure pen setting | 288 | Switch line thickness in display | 66 | Open Revit file | 16 |
| 21 | Open Revit File | 16 | Quit application | 39 | Configure pen setting | 288 |
| 15 | Render current view, photorealistic | 162 | Quit application | 39 | Open most recently used Revit file | 91 |
| 16 | Hide specified object category | 129 | Quit application | 39 | Undo | 7 |
| 17 | Render image file using cloud | 213 | Open Revit file | 16 | | |
| 19 | Activate design option | 78 | Minimize view | 315 | Delete component | 1 |
| 24 | Disjoint ends of structural element | 203 | Switch to default 3D view | 10 | Save current file | 12 |
| 25 | Orient camera to front | 144 | Switch to default 3D view | 10 | Save current file | 12 |

### 5.3.2 Organization-based analysis

The subsequent visual analysis at the organizational level further investigates the clusters. Figure 4 visualizes the ratio of clusters by type of organization. Architect firms and BIM operator firms have prominently high ratios of the dominant cluster relevant to their intensive modeling works. The architect firms held no minor clusters: potentially because the architect firms were in positions to initiate or issue design intent rather than coordinating with other models. Another tendency observed was that the division of Dominant clusters and Major clusters are equivalent in BIIM consultants and general contractors; it indicates a high proportion of non-modeling work. The university's logs have a noticeably large occupation of cluster #20 that successfully distinguished the command context from the practical projects.

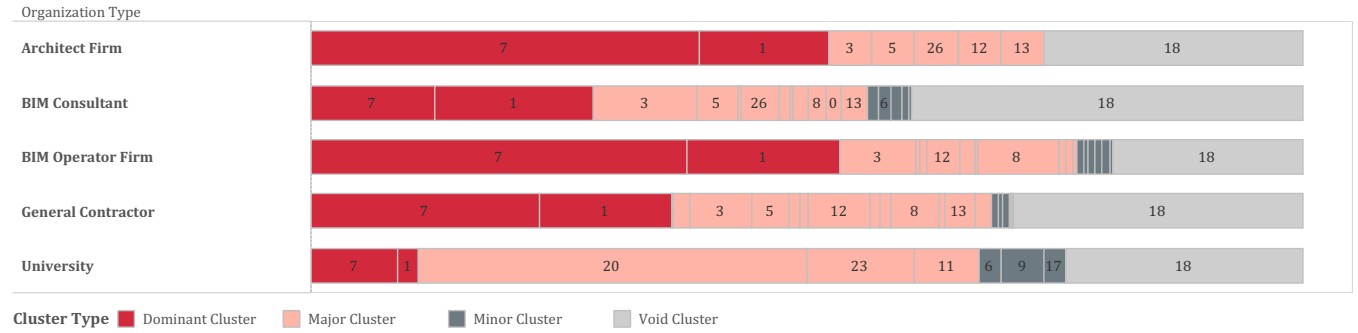

Figure 4. The ratio of clusters per type of organization.

Some organizations consisted of multiple data donors—the breakdown per data donor is separately presented in Figure 5. The size of dot plots reflects the percentage of overall logs from a donor.

The attribute of an organization often appears prominently in dominant clusters and major clusters; the log files of General Contractor A did not contain any of the dominant clusters. As confirmed in Table 2, this organization's usual BIM practice was receiving the model and applying it for construction. Cluster #6, which was represented by commands to measure dimensions, also reinforces that characteristic. Users exhibiting similar characteristics include user D from BIM Consultant G and user T from University D.

Clusters #20 and 23 appear almost exclusively in logs from University D. On the contrary, cluster #3, 8 contains logs for all organizations except university D, indicating that educational and industrial use of BIM was discriminated against. Investigating Cluster #27 contained in only General Contractor A and F's logs is expected to reveal the workflow specific to these organizations.

The activities among data donors are not inevitably similar within a single organization; they are considered strategically divided. For example, in BIM operator firm E, the logs from employees E, S, and Z are primarily classified into the dominant clusters. There are also users N, O, and Y, whose records are often classified into Minor clusters. Organization E was a corporation that produces project models under General Contractor F's direction. The former

BIM users concentrate on model production, while the latter audits and coordinates created models with others. On the other hand, the visualized activities of two users from General Contractor J resemble each other. The BIM tasks here are evenly shared and likely managed differently from organizations E and F.

**Predicted clusters per users, from organizations with multiple data donors**

|  | | BIM Consultant | | | | | | | | BIM Operator Firm | | | | | | | | | | General Contractor | | | | | | | | | | | | | | | University | | |
|---|---|---|---|---|---|---|---|---|---|---|---|---|---|---|---|---|---|---|---|---|---|---|---|---|---|---|---|---|---|---|---|---|---|---|---|---|
|  | | | | | | G | | | | | | | | E | | | | | | | A | | F | | | | H | | | | | J | | D | | |
| User name label | | 1 | 2 | 3 | 4 | A | C | D | L | E | F | H | N | O | Q | S | T | Y | Z | I | Y | A | I | S | Y | A | E | J | M | N | R | S | O | T | U |
| Dominant Cluster | 7 | | | | | | | | | | | | | | | | | | | | | | | | | | | | | | | | | | | |
|  | 1 | | | | | | | | | | | | | | | | | | | | | | | | | | | | | | | | | | | |
| Major Cluster | 3 | | | | | | | | | | | | | | | | | | | | | | | | | | | | | | | | | | | |
|  | 8 | | | | | | | | | | | | | | | | | | | | | | | | | | | | | | | | | | | |
|  | 12 | | | | | | | | | | | | | | | | | | | | | | | | | | | | | | | | | | | |
|  | 5 | | | | | | | | | | | | | | | | | | | | | | | | | | | | | | | | | | | |
|  | 13 | | | | | | | | | | | | | | | | | | | | | | | | | | | | | | | | | | | |
|  | 26 | | | | | | | | | | | | | | | | | | | | | | | | | | | | | | | | | | | |
|  | 20 | | | | | | | | | | | | | | | | | | | | | | | | | | | | | | | | | | | |
|  | 11 | | | | | | | | | | | | | | | | | | | | | | | | | | | | | | | | | | | |
|  | 14 | | | | | | | | | | | | | | | | | | | | | | | | | | | | | | | | | | | |
|  | 23 | | | | | | | | | | | | | | | | | | | | | | | | | | | | | | | | | | | |
|  | 0 | | | | | | | | | | | | | | | | | | | | | | | | | | | | | | | | | | | |
|  | 22 | | | | | | | | | | | | | | | | | | | | | | | | | | | | | | | | | | | |
|  | 27 | | | | | | | | | | | | | | | | | | | | | | | | | | | | | | | | | | | |
| Minor Cluster | 28 | | | | | | | | | | | | | | | | | | | | | | | | | | | | | | | | | | | |
|  | 6 | | | | | | | | | | | | | | | | | | | | | | | | | | | | | | | | | | | |
|  | 29 | | | | | | | | | | | | | | | | | | | | | | | | | | | | | | | | | | | |
|  | 4 | | | | | | | | | | | | | | | | | | | | | | | | | | | | | | | | | | | |
|  | 2 | | | | | | | | | | | | | | | | | | | | | | | | | | | | | | | | | | | |
|  | 9 | | | | | | | | | | | | | | | | | | | | | | | | | | | | | | | | | | | |
|  | 10 | | | | | | | | | | | | | | | | | | | | | | | | | | | | | | | | | | | |
|  | 21 | | | | | | | | | | | | | | | | | | | | | | | | | | | | | | | | | | | |
|  | 15 | | | | | | | | | | | | | | | | | | | | | | | | | | | | | | | | | | | |
|  | 16 | | | | | | | | | | | | | | | | | | | | | | | | | | | | | | | | | | | |
|  | 17 | | | | | | | | | | | | | | | | | | | | | | | | | | | | | | | | | | | |
|  | 19 | | | | | | | | | | | | | | | | | | | | | | | | | | | | | | | | | | | |
|  | 24 | | | | | | | | | | | | | | | | | | | | | | | | | | | | | | | | | | | |
|  | 25 | | | | | | | | | | | | | | | | | | | | | | | | | | | | | | | | | | | |
| Void Cluster | 18 | | | | | | | | | | | | | | | | | | | | | | | | | | | | | | | | | | | |

**Cluster Type**

- ■ Dominant Cluster   ■ Minor Cluster
- ■ Major Cluster   ■ Void Cluster

**% of Total Number of Records**

- 0.66%   40.00%   80.00%
- 20.00%   60.00%   100.00%

Figure 5. Clustered log file ratio per data donor under organizations.

### 5.3.3 User-based analysis

A series of visual analyses at the data donor level was summarized in Figure 6. Because the user survey is indispensable here to examine the discoveries, this section subjects only the correspondents to the user survey, unlike the preceding sections. The relevant questions in the questionnaire, listed in the Appendix, are indicated at the title of respective charts.

The cross-analysis with the users' discipline depicts that architect's logs are more likely clustered into dominant clusters than others. The logs from engineering (MEP) or others should comprise more non-modeling commands; their cluster distribution also confirms it.

Self-reported software skills articulate that users with proficiency left more dominant clusters and fewer void clusters. Although the proportion of dominant clusters with advanced skill was lower than that of the intermediate level, it becomes comparable when major clusters are jointed, seemingly because of the proficient users' manifold roles in projects.

The users' year of experience was irrelevant to the clustering results. Experienced

practitioners are generally apart from the BIM environment; however, such a tendency was not observed since the survey participants possessed a certain level of BIM skills.

The users with a lower percentage of working time in the software presented a prominently high portion of void clusters and less or even no dominant clusters. The least amount of void clusters was found in the group of users who spend 10-15% of their working time in Revit. Considered that more extended time in BIM software does not necessarily signify more efficient work, this observation explicitly depicts the significance of the proposed method, which can recognize the users more contributing to the project BIM progress yet not measurable by their working time or the level of self-evaluation.

**Predicted Clusters per Users' Discipline (Q19)**

**Predicted Clusters per Users' Self-evaluated Software Skill (Q3)**

**Predicted Clusters per Users' Year of Experience in the Industry (Q8)**

**Predicted Clusters per Users' Hourly Ratio Works on Revit (Q17)**

Cluster Type ■ Dominant Cluster ■ Major Cluster ■ Minor Cluster ■ Void Cluster

Figure 6. Cluster ratio, aggregated by users per answers of questions in the questionnaire.

### 5.3.4. Interpretation of cluster groups

The analyses so far enabled to abstract and interpret the clustering results. A two-dimensional plot can concisely organize the four cluster types, as shown in Figure 7, based on the number of logs classified and the model's contribution level.

The more logs categorized into the dominant cluster, the more intensive modeling work the user or group undertook. Dominant clusters become smaller when the organization is a receiver or coordinator of progressed models. The expanded non-modeling activities, particularly in minor clusters, signify the aligned team performance with the desired work scope.

In BIM team collaboration, management policies hugely vary depending on the strategy. The manager may evenly split the whole BIM task and share them among members or role-specifically assign to the specialized members. The proposed system helps monitor if the worksharing is as planned and enables teams and individual practitioners to self-monitor their BIM contribution.

At the individual level, a broad range of cluster types represents the user in charge of multidisciplinary, manifold tasks. Conversely, the same cluster repeatedly appears if the user

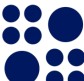

is specific task-oriented. This method allows both players and management to strategize in fulfilling expected BIM roles and enhancing the skills.

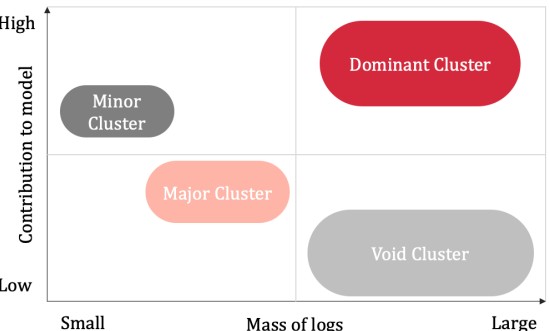

Figure 7. Abstract of cluster types from dataset 1.

## 6. Another case study

The exact process was applied to Dataset 2 as another implementation. Figure 8 displays an overview of clustering conforming to Figure 2 in the previous section.

As can be understood from Table 1, Dataset 2 has a relatively lower average number of executed commands per log file than Dataset 1. Reflecting this sparse data distribution, the number of clusters reduced to 20 despite the more immense log files.

The result displays that logs corresponding to the Void cluster (cluster #0) account for 60.9% of the total logs. As in the case study, we classified the cluster types into four categories: dominant cluster (#18), major clusters (#10, 1, and 3), and the others are minor clusters.

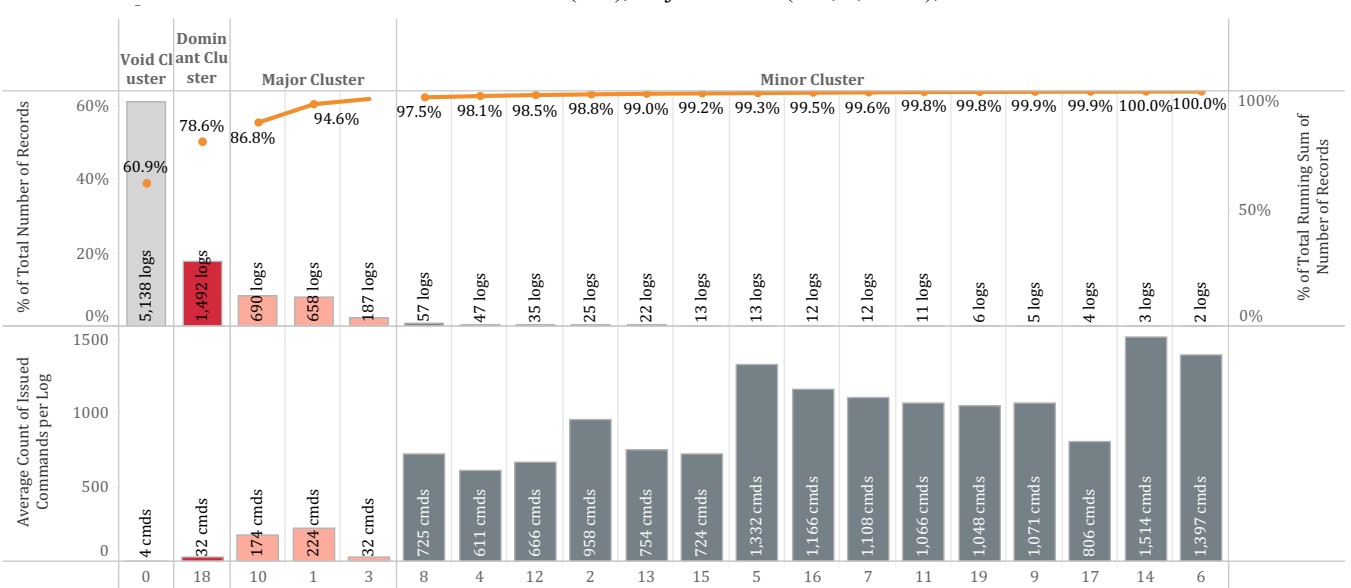

Figure 8. Overview of predicted clusters with four cluster types, dataset 2.

The observation leads to another allocation of four cluster types. The most typical cluster is still the Void cluster. However, the issued commands in the Dominant cluster averaged conspicuously fewer than in the rest. This fact amply proves that the principal BIM activity of these organizations is not placed on model creation but rather non-modeling activities. Also, unlike the case study, the average counts of commands of all minor clusters exceeded the major clusters' records. Minor clusters in this landscape proclaim a more outstanding contribution to the project model than major clusters.

Figure 9 displays the abstraction of the observed tendency in comparison with Figure 8. While it remains true that the dominant cluster signifies the representative BIM activity in the environment, more significant progress in project BIM was due to the major and minor clusters. It implies that these organizations will require BIM education for collaboration and management rather than the ordinary modeling-centric program. It is worth reiterating that the proposed method better fulfills the purpose of assessing the role and performance of BIM in these organizations than modeling efficiency measurement.

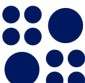

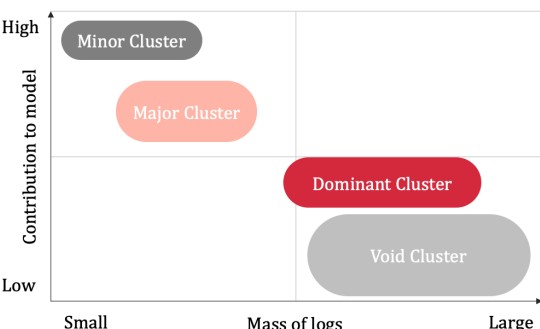

Figure 9. Abstract of cluster types from dataset 2.

## 7. Discussion

### 7.1. Proposed database

In both two different datasets, the most common log files were those classified as the void cluster. As these logs possess little information inside, they have not been highlighted in the existing BIM log mining approaches focusing on model progress. Dataset 2 includes the data donors not deeply engaged with BIM workflow. Indeed, there were fifty-two users for whom all log files were classified into the Void cluster, accounting for 28.5% of all participants (Figure 10). While some of these users are potentially unpracticed and experiencing difficulty with the BIM, they are equally likely qualified designers who make crucial decisions by viewing the model. They may also be ingenious detailers who actively develop design by two-dimensional CAD drawings along with BIM progress. Analyses targeting only the predominant BIM users may have a risk of disregarding those users' presence. The partitioning concept and large clustering do not filter such information; instead, they amplify those weak signals to draw enough attention to non-modeling-related activities. Hence, this holistic approach enables more inclusive analysis, especially for large-scale projects or multidisciplinary project orchestration.

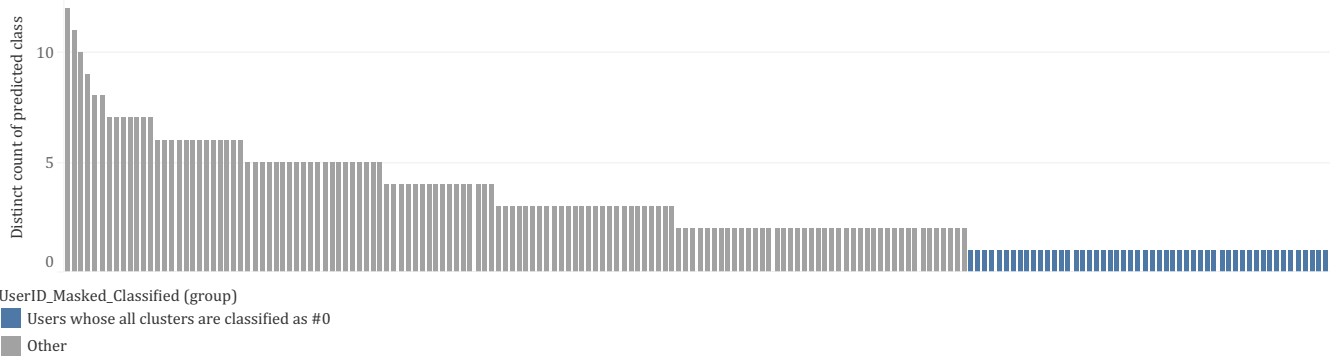

Figure 10. Distinct count of predicted class per user, dataset 2.

In the case study, the minor clusters consisted of commands rarely issued. Those commands often have distinguished functionality, hence lead to very contributing activities. Unlike modeling events that appear dozens of times, such commands are executed mostly once in the software session. The advantage of machine learning algorithms lies in capturing those offset signals and reflecting them in the classification. Thus, the application of classification strategy suits intending to capture the varied activities holistically and inclusively.

The case study demonstrated the idea to overview the analysis by four cluster types and their ratios at the organizational level (both by type and individual) and user level. It can likewise be applied at the group or project levels as well. This scalability also proves the significance of the proposed methodology. For example, the project BIM team's performance under a general contractor can be analyzed side-by-side against a BIM specialist company's group. The BIM collaboration in specific building types, such as healthcare or transportation facilities, can be portrayed and benchmarked with other successful past projects. Such flexibility is advantageous to comprehend the dynamic and diverse BIM collaboration landscape.

Interpreting visualized analysis does not demand extensive BIM software skills. It requires insight into AEC's projects and teamwork more than ever. As the management layers of projects

and firms have often had limited practical BIM knowledge, the individual interview was almost the only way to investigate the real collaboration. The proposed process provides instant and detailed visualization of the BIM ecosystem. Nevertheless, this data-driven management does not exist for the sake of micromanagement. It opens up the implicit process to active communication for operational improvement through a dynamic, universally understandable method. Such an approach has been conventionally possible by the power of outstanding BIM managers; this study aims to democratize such tacit know-how for open and ad-hoc BIM management.

### 7.2. Significance of the proposed process

Dataset 1 explicated that the comprehension of clustered BIM log can be enhanced when matched with information about the data donors and their organizations from three perspectives: command level, organizational level, and user level. The readability of the visualized BIM activities has dramatically improved by the developed methodology. Monitoring the practitioner or firm at the organizational, project, team, and individual levels grants multiple benefits to their own, including 1) the effect assessment of BIM training program, 2) improved staffing and team building, 3) the skill design by benchmarking against others.

Dataset 2 confirmed the applicability of the proposed process. However, the four cluster types were translated differently. This contrasted interpretation arises from different ecosystems depending on the project region, type, size, et Cetra.

A project dilemma exists that the expected BIM benefit in a large project involving many stakeholders is significant, yet the project's uniqueness makes it more challenging to implement BIM. The BIM implementation in international projects tends to stall; the quality of the final output can deteriorate. Even worse, the use of BIM itself becomes abandoned amid the project.

To improve the situation, it is pivotal that organizations and project teams identify their own organizations' traits. It further encourages collaboration within projects, which is the mainstay of BIM. Ultimately, the desirable lifecycle will be realized where BIM utilization will be maintained from design to construction and will be part of the solution for further data utilization.

## 8. Limitation of study and future prospect

The log files composed a comprehensive database, which was vital for the machine-learning algorithm to solidify the clustering approach. Increasing the number of log files enriches the database. The accumulated log files can transform into teaching data for supervised machine-learning, by which the results over different projects and organizations became comparable. Further research is encouraged to enhance the quantity of data from large-scale projects to solidify the findings and discover more details of non-modeling activities.

Since the number of clusters is non-deterministic in the clustering problem, the adequate number of clusters is not self-evident and must be determined by study results. The number adopted in this study is consistent with the objective of providing a broad overview of activities consisting of multiple BIM organizations. However, the optimal number of clusters needs to be further examined.

The research began with a collection of BIM log files from as broad as possible organization types. On the other hand, it was very challenging to obtain enough data across organizations. Many organizations declined to share their log files due to project confidentiality. For example, some components, such as a radiation generator or a baggage handling system, can hint at the project types or even the client itself. The log files contain hardware information as well, which some firms considered confidential. Deleting those kinds of information at the data collection stage is desirable; however, this requires running such systems at the data donor's premises. Gathering a plentiful quantity of log files from a limited group of organizations will lead to cumulative insights for the topic.

Lastly, a limitation of this approach is that the log data itself originates from a single software program. The integration of structural analysis or special equipment engineering will not happen even for central BIM authoring software such as Revit; BIM must be an open platform to accept the interoperation of multiple professional programs [53]. For software programs that do not leave usable logs, there needs to be a supplemental system that can collect the operational data. Yarmohammadi et al. leveraged an application programming interface (API) to collect the modeling data in real-time from Autodesk Revit [54]. A similar approach is possible when the software vendor provides such an environment. Expanding the range of target software will further contribute to the body of knowledge in this domain.

## 9. Conclusion

A novel process was devised to enhance the existing BIM log mining approach. Visual analytics stands on four broad categories of BIM activities by the clustering algorithm. Case studies confirmed the usefulness of the method in deciphering BIM activities at the levels of commands, organizations, and users. Intercomparison of BIM activities became possible across the organizational, project, and individual levels. The comprehensive visualization does not require BIM expertise to examine.

Due to various factors such as organizational disciplines, project scale, and the project requirement, diverse BIM capabilities are required for BIM users and teams. As the service they provide to the project changes dynamically along the project timeline, their formations in the environment should change from time to time as needed. The approach proposed in this study is not a metric for assessing individual or group performance but rather a tool for reflecting whether the desired skills and competencies are being provided by benchmarking and comparing others or past selves. The method enables BIM practitioners to design their skills and provide BIM teams with information for their organizational design.

The successful BIM implementation throughout the building lifecycle is still hard to realize. Considering the organizational design and individuals' skills are deeply relevant to the issue, this methodology potentially becomes an indispensable protocol to continuously improve the use of BIM.

### Acknowledgment

The authors herein express sincere appreciation to the data donors across the donor organizations and countries. This research was not possible without Mr. Soki Nakamura's advice on machine-learning applications. Professor Yasushi Kiyoki introduced the idea for the semantic space, which drastically streamlined the research flow.

This research did not receive any specific grant from funding agencies in public, commercial, or not-for-profit sectors.

### Declaration of competing interests

The authors declare that they have no known competing financial interests or personal relationships that could have appeared to influence the work reported in this paper.

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

## Appendix: user survey questions

The following questions were used for the study.

Q1   When did you first use Revit by yourself?

Q2   When did you start using Revit for your ongoing project?

Q3   How would you rank your own Revit skill within your firm/organization?
++++ Expert / +++ Advanced / ++ Intermediate / + Basic / Beginner

Q4   Please choose the most helpful resource during your Revit learning process.
Online help, manual, video / Online discussion board, SNS / Book, publication / Asking colleague or friend / Trial and error / Vendor's help desk

Q5   In your average daily work, how much time do you spend on operational troubleshooting?
Almost none / 30 minutes / One hour / One and a half hour / More than two hours

Q6   How would you evaluate your BIM skill as compared with CAD?
Better at BIM / Almost equal / Better at CAD / Neither

Q7   Which of the following best applies to your position regarding decision making?
NA: I am a student or trainee / NA: I am a consultant / I model with the given instructions / I initiate, but it is subject to supervisor's approval / I am the decision-maker

Q8   How many years of experience do you have in the AEC industry (including civil works)?

Q9   How many years have you been in your current company/college/organization?

Q10   Which output do you produce the most in BIM or in work based on BIM geometry?
Design drawings / Detail drawings / Construction, shop drawings / Perspective images / Navisworks model / Revit model

Q11   In terms of your project involvement, which of the following actions best match your role?
Creating models / Updating or amending others' models / Auditing models / Extracting information out of models / Using models for reference / Coding on models

Q12   In your recent projects, which status best applies to the project BIM model versus project drawings?
Model is the most updated / model is as updated as drawings / Model is the most updated, drawings partially up to date / Drawing is the most updated, the model is catching up / Drawing is the most updated, the model is for reference / Model is abandoned / Using the model as a test case

Q13   Which of the following best applies to the models in your recent projects?
I work on a single model per project / I work on a model in federated models / I coordinate multiple models

Q14   Which description best describes the model(s) you work on?

Models of the project that I am in charge of / Models of the project that I provide BIM support for / Miscellaneous models of your firm/organization's projects / Miscellaneous models, various authors / Non-project, sandbox models

Q15  Please choose the task that takes the longest time to perform in Revit.
Model creation / Documentation / Model amendment / Model checking / Rendering, visualization / Data conversion

Q16  Do you use Dynamo in your project?
No, I have not used Dynamo / Yes, I leverage codes made by others / Yes, I code by myself / Yes, I create codes with programmers

Q17  About what percentage of your time working in Revit do you stay in 3D views?
0-5% / 5%-10% / 10%-15% / 15%-20% / 20%-33% / 33%-50% / >50%

Q18  How has your experience been with Revit?
5: I strongly like it / 4: I like it / 3: Neutral / 2: I dislike it / 1: I strongly dislike it

Q19  Which of the following is your discipline?
Architecture / Structure / MEP / Construction / Others

Q20  Please select your job title.
CEO, President / Manager, Director / Designer, Engineer / Modeler, Coordinator / Consultant / Student, Trainee

抄録

本論文では Building Information Modeling ソフトウエアのログ解析手法（BIM log mining）の拡張を提案する。複数の組織から収集したログファイルを、記録されているコマンドに基づきクラスタ化し、ビジュアル分析を行った。ケーススタディとして2つの異なるデータセットを異なるレベル（コマンド単位、組織単位、ユーザ単位）で分析し、多様な BIM 活動が理解しやすく可視化できることを示した。結果、実行されたコマンド同士の共通性・プロジェクトに対する貢献度に対応してログファイルは 4 種類に大別できることが判明した。この分類によれば複雑で見えにくい BIM 上での活動を、予備知識を必要とせずに解読できる。その有用性は相互比較、とくに組織の業態や規模を超えたスケールフリーな比較にあり、データに基づいた個人や組織のスキルデザインを可能にする。個々のユーザーや BIM チームは、本手法により動的に活動をモニタリングし、時々刻々と変化するプロジェクトの状況によりよく対応することができる。BIM はプロジェクト期間中に活用が停滞しやすいことが知られている。本手法はスキルや管理の問題を改善することで竣工時まで BIM が運用されやすい状況をつくり、特に大規模プロジェクトで期待される BIM 利活用の効果向上に貢献する。

# Response letter

Manuscript ID：21A001

Paper Title: Visual log analysis method for designing individual and organizational BIM skill
Authors: Tsukasa ISHIZAWA and Yasushi IKEDA

| Reviewer No. | Fill in page lines, figures, tables, etc. | | Reviewer's comments ※Do not omit any comments. | Response (Purpose of the revision, author's opinion, etc.) |
|---|---|---|---|---|
| | Original | Revised | | |
| 1 | Fig.1 | P3, L9 | This is an inaccurate and misleading conceptual diagram.
• If this diagram shows the generic condition of Japanese BIM-based projects, please explain why it includes the "University." If this diagram indicates the specific condition for this paper, it has to be located in the appropriate section with additional explanation.
• Please indicate the references for the diagram showing "organizational size" and "project timeline."
• Please indicate the references for the diagram "BIM functionality."
• Please explain the meaning and validity of the thickness of the line connecting "the diagram showing the organizational size and project timeline" with "the diagram of BIM functionality." | The diagram intended to elaborate on the complex relationships between the BIM users and the software functions. The figure was removed as the above was literally explained in Chapter 2. |
| 1 | P3, L31 | P3, L31 | Scale-free intercomparison
• Please specify what scale-free intercomparison is?
• Is it possible to achieve a scale-free comparison? How? | The term "scale-free" was to claim that the proposed system enables the intercomparison of characteristics in BIM use across projects, organizations, and users. The relevant phrase was rectified as above. |
| 1 | P6, L13 | P6, L2 | The design and construction processes vary in different countries, therefore assigned job titles to have different roles and responsibilities. For example, a general contractor in Japan includes a design department (people who play the roles of architects and engineers), so the organizational structure and role are different from that of a general contractor in Singapore or Slovakia. It is inappropriate to lump them all together and call them "general contractors." This point also applies to Figure 1, 5.3.2 Organization-based analysis, and 5.3.3 User-based analysis.
• Please explain and define the "type of organization" more precisely, including "General Contractor," "BIM Consultant," "BIM Operator firm."
• Please revalidate the results of the analysis and rearrange the tables, if necessary. | The scope of services of organizations are additionally explained. It was also clarified that the general contractor A, F, H, and J provides the design-build services to distinguish them from builders. These organizational backgrounds are already taken into account. |

| 1 | N/A | P6, L11 P15, L42 | Please explain the appropriateness of the dataset obtained. 
 • Please explain that the obtained dataset is sufficient to derive the results. 
 • If there is any bias in the dataset obtained, please specify what difference it might make in the analysis results. | As stated in 5.1, The number of key factors (issued commands) was 622,273, which targeted to draw parallels to the existing literature (181,969 to 620,492). 
 The authors attempted to eliminate the bias in datasets, especially for dataset 1; however, many companies declined to share their log files. Enriching the data is expected in future research. These limitations are stated in section 8. |
|---|---|---|---|---|
| 1 | Abstract | P2, L16 P3, L31 P15, L1 | This paper proposes the methodology to decipher recorded BIM activities "without requiring background knowledge." Please explain what kind of person referred to. What kind of background knowledge is required or not required. | As the title mentions, the method is for individuals and organizations to design their BIM skills. Past studies clarified that the project managers, in general, have minimal knowledge of BIM (1.2). Thus the performance monitoring systems were devised to promote the use. Instead of the BIM software skill, the insight into AEC projects and teamwork (7.1). 
 A further statement is added in section 2 to clarify the expected users of the proposed system. |
| 1 | N/A | P14, L33 | The different building types, such as repetitive spaces like schools and hospitals, and unique spaces like a museum, may have different BIM operations even in the same project phase or design organization. Is this influence the results? If so, how? | BIM workflows can vary by many factors, including project size, building use, contractual condition, and local commercial customs. The proposed method enables the flexible visualization of BIM contribution regardless of those conditions specific to projects. Increasing the data amount will further strengthen the benefit. |
| 2 | P3, L31 | P3, L31 | In complex network theory, "scale-free" means the power law of the degree distribution. What is "scale-free intercomparison" in this paper? | The term "scale-free" was chosen to claim that the proposed system enables the intercomparison of characteristics in BIM use across projects, organizations, and users. The statement was revised to avoid confusion. |
| 2 | P5, L14 – 21 | P4, L58 | The meaning of S15 and S13 is difficult to understand only with this context. Isn't this paragraph unnecessary because it only explains the basic etiquette of the academic paper structure? | The paragraph was revised to minimally state the structure and application of Design Science Research Methodology. |
| 2 | P7, L36 | P7, L20 | Please confirm that the number of objects in each cluster is not always equal in the K-means method. | The sentence states that the cluster sizes are *implicitly assumed* to be equal. The result displays that the cluster sizes vary. |

| 2 | P6, 5.2.2 | P6, L25 | It is not clearly explained what were used as feature values to classify the journal files. Are they the number of times 798 different commands are executed? | The classification aimed to separate the log files based on the types of issued commands and their frequency during a single software session. The clustering process requires the structured aggregation of the issued command types per user. Sentences were added to clarify above. |
|---|---|---|---|---|
| 2 | P8, L6 | P8, L5 | Has it been verified that the separation result in terms of the number of log files generally "follows the power law"? | No, it has not. The relevant statement was removed since such a tendency is not critical for the subsequent analysis. |
| 2 | P8, L26 | P8, L29 | Criteria for separating Major clusters and Minor clusters are difficult to understand. | The split of Major clusters and Minor clusters are conducted at the double standard deviation in the running total, considering the occurrence which may have influenced the significance in model contribution. The appropriateness of the division was tested through the following visual analytics. |
| 2 | P7, 5.2.2 | P4, L13

P15, L39 | What are the criteria for dividing into four groups? Why didn't you employ hierarchical clustering methods from the beginning though you will group clusters later? As a result, there is a possibility that it will not necessarily be four. | The implication of clusters was subsequently discovered through the following analyses. As stated in section 8, the number of clusters must be determined by study results; thus, the number of super groups may vary in another dataset: for example, log files explicitly from skilled and tacit modelers. The discovery of the research is not about the number of groups but the method to decipher the unstructured BIM log to harvest insight about the contribution to the project. |
| 2 | P9, L19 | P9, L19 | No command with a blue background is found in Table 5. | The statement was irrelevant. The sentence is removed. |
| 2 | P10, L7 – 8 | P10, L9 | As far as Figure 5 is concerned, the proportions of Dominant clusters of BIM consultants and general contractors appear to be different. | The division of Dominant clusters and Major clusters are equivalent in BIM consultants and general contractors. The statement has been rectified as above. |
| 2 | P11, Figure 6 | P11, Figure 5 | Why are numbers and alphabets mixed in donor IDs? | They are the identifier label of data donors. Numbers signify anonymous donors from the organization. |
| 2 | P10, L17 – 19 | P10, L23 | User C does not contain # 6 logs | It has been amended as User D. |
| 2 | P12, Figure 7 | P12, Figure 6 | Figure 7 should correspond to the question number in the appendix. | The relevant numbers of questions in the appendix were added at the title of respective charts. |

| 2 | P12, L10 | P12, L9 | How can we recognize the users undertaking critical roles only with the proposed method? | To be precise to the intended meaning, the phrase was rectified as "more contributing to the project BIM progress." |
|---|---|---|---|---|
| 2 | P11 – P12 | P7, L33 | In 5.2.2, it is stated that the results of the user survey were used as ground truth to validate the analysis results. Generally, validation in unsupervised machine learning is whether the result of clustering can reproduce known classifications. According to this understanding, it is unclear what was "validated" in 5.3.3. | The statement aimed to explain that the clustering results are to be examined through visual analytics by incorporating the user survey. The relevant sentence was rectified accordingly. |
| 2 | P13, L7 - 10 | P13, L7 – 10 | The average value of the command would be the value including the Void cluster. Since Figure 9 gives an erroneous impression, it is better to clearly indicate "Table 1 on Page 6". These paragraphs should also be moved before Figure 9. | It was understood that Figure 8 (previously 9) is confusing with Figure 3 (2), not Table 1. The expression was amended, and the paragraphs are relocated before Figure 8 (9). |
| 2 | P13 | P13 Chapter 6 | Chapter 6 merely carried out the same analysis process using different data, and it is difficult to said that the process was verified. | Chapter 6 was to validate the utility of the proposed method over the alternative dataset. To avoid miscommunication, the section and relevant phrases are rectified as "another case study." |
| 2 | P14, L6 - 19 | P14 L6 – 19 | The authors mentioned the important point in this paragraph. It is a pity that the possibility pointed out cannot be verified because it is difficult to identify by the proposal process alone. | At the moment, the possibility pointed out in the paragraph only remains as the discussion. For example, detailed interviews with the candidates will be effective in verifying the hypothesis. It is carried out and published in the author's subsequent paper. |