# OpenReview forum: "Visual log analysis method for designing individual and organizational BIM skill"
_AIS-J.org/2021/Journal_

### Official Review · AnonReviewer2 · 2021-08-16

**Rating:** 2
**Confidence:** 3

**Review:**

P3, L31 : In complex network theory, “scale-free” means the power law of the degree distribution. What is "scale-free intercomparison" in this paper?

P5, L14 – 21 : The meaning of S15 and S13 is difficult to understand only with this context. Isn't this paragraph unnecessary because it only explains the basic etiquette of the academic paper structure?

P7, L36 : Please confirm that the number of objects in each cluster is not always equal in the K-means method.

P6, 5.2.2 : It is not clearly explained what were used as feature values to classify the journal files. Are they the number of times 798 different commands are executed?

P8, L6 : Has it been verified that the separation result in terms of the number of log files generally "follows the power law"?

P8, L26 : Criteria for separating Major clusters and Minor clusters are difficult to understand.

P7, 5.2.2 : What are the criteria for dividing into four groups? Why didn't you employ hierarchical clustering methods from the beginning though you will group clusters later? As a result, there is a possibility that it will not necessarily be four.

P9, L19 : No command with a blue background is found in Table 5.

P10, L7 – 8 : As far as Figure 5 is concerned, the proportions of Dominant clusters of BIM consultants and general contractors appear to be different.

P11, Figure 6 : Why are numbers and alphabets mixed in donor IDs?

P10, L17 – 19 : User C does not contain # 6 logs

P12 : Figure 7 should correspond to the question number in the appendix.

P12, L10 : How can we recognize the users undertaking critical roles only with the proposed method?

P11 – P12 : In 5.2.2, it is stated that the results of the user survey were used as ground truth to validate the analysis results. Generally, validation in unsupervised machine learning is whether the result of clustering can reproduce known classifications. According to this understanding, it is unclear what was “validated” in 5.3.3.

P13, L7 - 10 : The average value of the command would be the value including the Void cluster. Since Figure 9 gives an erroneous impression, it is better to clearly indicate "Table 1 on Page 6". These paragraphs should also be moved before Figure 9.

P13 : Chapter 6 merely carried out the same analysis process using different data, and it is difficult to said that the process was verified.

P14, L6 - 19 : The authors mentioned the important point in this paragraph. It is a pity that the possibility pointed out cannot be verified because it is difficult to identify by the proposal process alone.


**Overview:**

It is highly commendable that the previous research is widely referred to, the original data is carefully analyzed, and the paper is written in great detail. However, there are some unknown points in the important part.

---

### Official Review · AnonReviewer1 · 2021-08-22

**Rating:** 2
**Confidence:** 3

**Review:**

2．Research question and expected contribution

Fig.1
This is an inaccurate and misleading conceptual diagram.
-	If this diagram shows the generic condition of Japanese BIM-based projects, please explain why it includes the "University." If this diagram indicates the specific condition for this paper, it has to be located in the appropriate section with additional explanation.
-	Please indicate the references for the diagram showing "organizational size" and "project timeline."
-	Please indicate the references for the diagram "BIM functionality."
-	Please explain the meaning and validity of the thickness of the line connecting "the diagram showing the organizational size and project timeline" with "the diagram of BIM functionality."

Scale-free intercomparison
-	Please specify what scale-free intercomparison is?
-	Is it possible to achieve a scale-free comparison? How?

5.1 Data Collection
The design and construction processes vary in different countries, therefore assigned job titles to have different roles and responsibilities. For example, a general contractor in Japan includes a design department (people who play the roles of architects and engineers), so the organizational structure and role are different from that of a general contractor in Singapore or Slovakia. It is inappropriate to lump them all together and call them "general contractors." This point also applies to Figure 1, 5.3.2 Organization-based analysis, and 5.3.3 User-based analysis.
-	Please explain and define the "type of organization" more precisely, including "General Contractor," "BIM Consultant," "BIM Operator firm."
-	Please revalidate the results of the analysis and rearrange the tables, if necessary.

5.2　Development of algorithm
Please explain the appropriateness of the dataset obtained.
-	Please explain that the obtained dataset is sufficient to derive the results.
-	If there is any bias in the dataset obtained, please specify what difference it might make in the analysis results.

Others
-	This paper proposes the methodology to decipher recorded BIM activities "without requiring background knowledge." Please explain what kind of person referred to. What kind of background knowledge is required or not required.
-	The different building types, such as repetitive spaces like schools and hospitals, and unique spaces like a museum, may have different BIM operations even in the same project phase or design organization. Is this influence the results? If so, how?


**Overview:**

The paper analyzes the BIM skills of individuals and organizations using log mining methods. This method provides deep insight into the project management of BIM utilized projects. However, there are some inadequacies and incompleteness in the study, as shown below. Please make up for these inadequacies and complete the paper.